# EFFICIENT OBJECT-CENTRIC LEARNING FOR VIDEOS

## ABSTRACT

This paper introduces a method for efficiently learning video-level object-centric representations by bootstrapping off a pre-trained image backbone, which we term `Interpreter`. It presents a novel hierarchical slot attention architecture with local learning and an optimal transport objective that yields fully unsupervised video segmentation. We first learn to compress images into image-level object-centric representations. `Interpreter` then learns to compress and reconstruct the object-centric representations for each frame across a video, allowing us to circumvent the costly process of reconstructing full frame feature maps. Unlike prior work, this allows us to scale to significantly longer videos without resorting to chunking videos into segments and matching between them. To deal with the unordered nature of object-centric representations, we employ Sinkhorn divergence, a relaxed optimal transport objective, to compute the distance between unordered sets of representations. We evaluate the resulting segmentation maps on video instance segmentation in both realistic and synthetic settings, using YTVIS-19 and MOVi-E, respectively. `Interpreter` achieves state-of-the-art results on the realistic YTVIS-19 dataset and presents a promising approach of scaling object-centric representation learning to longer videos.[1]

## 1 INTRODUCTION

Self-supervised object-centric learning has received significant attention within the last few years (Locatello et al., 2020; Aydemir et al., 2023; Kakogeorgiou et al., 2024). In object-centric learning the objective is to learn a mapping from observations to sets of vectors such that each vector encodes a semantically distinct part of the observation, e.g. individual or groups of objects in an image. This process is learned in a self-supervised manner, for example through reconstruction of image pixels or feature maps. Recent work has extended object-centric learning from images to video while achieving impressive results on video object segmentation without relying on additional cues such as motion estimation or depth. However, extending slot attention to videos, particularly of high resolution, presents a computational challenge due to the requirement of reconstructing the full feature map of each frame. As a result, prior work (Aydemir et al., 2023) resorts to chunking videos into small segments of a few frames, extracting object-centric representations and segmentations, and matching these across frames. We submit that a more efficient method is needed to permit full processing of the temporal context.

We present one such solution. `Interpreter` introduces a novel hierarchical slot attention approach with a local learning objective. We overcome the temporal limitations of prior work by breaking the problem down into two parts. `Interpreter` first learns to compress image feature maps into object-centric representations through reconstruction. Next, our proposed method learns to compress and reconstruct the image-level object-centric representations across *entire* videos, yielding a set of video-level frames. However, to implement this approach requires us to answer a few questions. The first question is how to extend the attention maps between the individual frames and video-level representations. For this we show that attention maps can be *propagated* between the image and video level slot attention modules to compute a image-to-video attention maps. The second question is what criteria is suitable to measure the distance between the unordered sets that constitute the image-level object-centric representations and their reconstructions. This we resolve by considering any two sets of object-centric representations as empirical samples in high dimensional

---

[1]Code to be made publicly available at a later date.

feature space. Through this we are able to employ Sinkhorn Divergence, an entropy regularized optimal transport distance that is amenable to gradient-based learning.

To summarize, our contributions are threefold. **First**, we introduce `Interpreter`, a novel method that leverages a hierarchical slot attention architecture with local learning objectives to efficiently learn video-level representations from pre-trained image backbones, enabling the processing of entire videos without the need for chunking. **Second**, we propose a technique to propagate attention maps between image-level and video-level representations, facilitating the computation of image-to-video level attention maps. **Third**, we address the challenge of measuring distances between unordered sets of object-centric representations by employing Sinkhorn Divergence, an entropy-regularized optimal transport distance. We validate our approach on both realistic and synthetic datasets, demonstrating state-of-the-art video object segmentation performance on YTVIS-19 in terms of mean Intersection over Union.

## 2 RELATED WORK

We consider here two lines of work, object-centric learning and video object segmentation, that we will summarize in brief. We focus on fully unsupervised video object segmentation.

**Object-Centric Learning:** The most prevalent method in contemporary object-centric learning is Slot Attention (Locatello et al., 2020), which defines a learnable iterative function that extracts a set of slots of predefined cardinality from an observation, e.g. an image, through an inverted cross-attention mechanism. Subsequent work extended Slot Attention from simple pixel reconstruction on synthetic scenes to real world images by introducing a feature map reconstructive objective using the high quality features from a pre-trained DINO/DINOv2 ViT backbone (Seitzer et al., 2022; Caron et al., 2021; Oquab et al., 2023). Newer versions of Slot Attention have also been introduced, such as Invariant Slot Attention (Biza et al., 2023), which makes slot attention invariant to translation and scale, and Implicit Slot Attention (Chang et al., 2022) which significantly improved the robustness of training.

**Video Object Segmentation:** In Video Object Segmentation (VOS) the goal is to accurately segment individual objects throughout the frames of a video. This includes making sure that segmentations stay consistent across the sequence, without mix up of identities. While there exist two common VOS settings, semi-supervised and unsupervised, we choose here to focus on prior work in unsupervised VOS as it aligns with our proposed method. In unsupervised VOS training on ground truth segmentation annotations is permitted, but no supervision is provided at inference time. Recent work, however, has approached VOS *fully* unsupervised (Xie et al., 2022; Aydemir et al., 2023; Ding et al., 2024), where the only permitted supervision should come from the data itself. Methods such as OCLR (Xie et al., 2022) use optical flow to discover and segment objects. More recently, approaches not reliant on any modality other than the video itself, have been proposed. Two notable examples are SOLV and BA (Aydemir et al., 2023; Ding et al., 2024).

SOLV combines per-frame invariant slot attention with learnable queries and a temporal encoder that enriches slots across frames. SOLV also introduces agglomerative clustering to alleviate the over-clustering that occurs from using a fixed number of slots. BA takes a different approach and makes use of the attention maps of a pre-trained DINO vision encoder and learns an attention mechanism to capture spatio-temporal dependencies, followed by hierarchical clustering to generate coherent object segmentation masks. However, both SOLV and BA are limited by relatively short temporal contexts, e.g. T = 5 and T = 3 frames, respectively, requiring matching between video segments. In contrast, our proposed method, `Interpreter`, is designed to allow for a more extensive temporal context, making it more effective at maintaining object consistency across longer sequences.

## 3 METHODOLOGY

In this section we introduce the architecture and training objective of `Interpreter`. We also describe how to propagate the attention maps between image features and video-level representations.

## 3.1 INTERPRETER

Our proposed method, `Interpreter`, follows prior works (Seitzer et al., 2022; Kakogeorgiou et al., 2024) by first learning an image-level slot attention auto-encoder to reconstruct high-quality image features from a pre-trained DINOv2 ViT-B/14 backbone. Subsequently, we introduce a second video-level slot attention module that is trained by reconstructing the image-level object-centric representations from the first level. However, the video-level objective poses a challenge. While the image-level reconstruction benefits from an explicit ordered structure in the feature map, allowing for structured losses like L2, the image-level object-centric representations form an *unordered set*, thereby *lacking explicit structure*. Although these representations implicitly contain latent positional information, they still do not permit the use of a structured loss.

To effectively use the object-centric representations as reconstruction targets, we require a loss function that is invariant to the ordering of elements within the set. Consider a set of object-centric representations $\mathcal{S}$ and a predicted set $\tilde{\mathcal{S}}$, both viewed as empirical samples in high-dimensional feature space with equal probability mass assigned to each feature vector. One natural way to measure the discrepancy between $\mathcal{S}$ and $\tilde{\mathcal{S}}$ is the Wasserstein (optimal transport) distance, which evaluates the minimal "work" (distance × mass) required to transform $\tilde{\mathcal{S}}$ into $\mathcal{S}$. This distance is inherently invariant to the ordering of elements. However, the direct computation of optimal transport is non-differentiable, complicating its use in gradient-based optimization.

Fortunately, an alternative divergence measure, **Sinkhorn Divergence**, approximates the Wasserstein distance while being both fast and differentiable (Feydy et al., 2019). Sinkhorn divergence interpolates between the exact Wasserstein distance and an entropy-regularized divergence by incorporating an entropy term that smooths the problem. This divergence can be computed iteratively via the Sinkhorn-Knopp algorithm. As the entropy regularization diminishes, Sinkhorn divergence converges towards the true Wasserstein distance, making it a practical and theoretically sound choice for measuring how well the model reconstructs unordered sets in high-dimensional feature space.

## 3.2 ARCHITECTURE AND OBJECTIVE

`Interpreter` consists of a pre-trained image backbone and two hierarchically stacked slot attention auto-encoders that are trained with a local objective, independently. For each level we use Implicit Slot Attention (Chang et al., 2022) for more robust training and Flash Attention (Dao et al., 2022) for faster and cheaper transformer attention (Vaswani et al., 2017). For calculating the Sinkhorn divergence we use the GPU accelerated `Geomloss` (Feydy et al., 2019) package due to its speed and robust implementation.

**The first level** of the hierarchy consists of an slot attention module and a transformer decoder without causal masking. A diagram of this stage of training can be seen in figure 1 left. The slot attention module takes image feature maps $F \in \mathbb{R}^{HW \times D_0}$, from the pre-trained image encoder and compresses them to object-centric representations $\mathcal{S} \in \mathbb{R}^{N \times D_1}$, where $N$ is the number of image slots with $N \ll HW$, $D_0$ and $D_1$ is the dimensionality of the feature map and slots, respectively. Following this, a transformer decoder maps a set of 2D sine-cosine positionally encoded vectors $P_0 \in \mathbb{R}^{HW \times D_0}$ to a reconstructed feature map $\tilde{F}$ conditioned on the slots $S$. The objective is then calculated as the L2 loss between the original feature map $F$ and the reconstructed feature map $\tilde{F}$:

$$\mathcal{L}_0 = \|\tilde{F} - F\|_2^2 \tag{1}$$

**The second level** of the hierarchy consists of a transformer encoder, a slot attention module and a transformer decoder, also without causal masking. A diagram of this stage of training can be seen in figure 1 right. The slots $\mathcal{S}_0, \ldots \mathcal{S}_T$ for $T$ frames from the prior level are first concatenated and time-positionally encoded with 1D sine-cosine positional encodings, after which they run through the transformer encoder to produce temporally enriched representations. The temporally enriched image slots are then compressed into a set of video-level slots $\hat{\mathcal{S}} \in \mathbb{R}^{K \times D_2}$ using the slot attention module, where $K$ is the number of video slots with $K \ll TN$ and $D_2$ is the video slot dimensionality. Similar to (Aydemir et al., 2023), we employ agglomerative clustering with complete linkage using cosine distance as the criterion to merge slots after slot attention, to mitigate over-clustering. Finally,

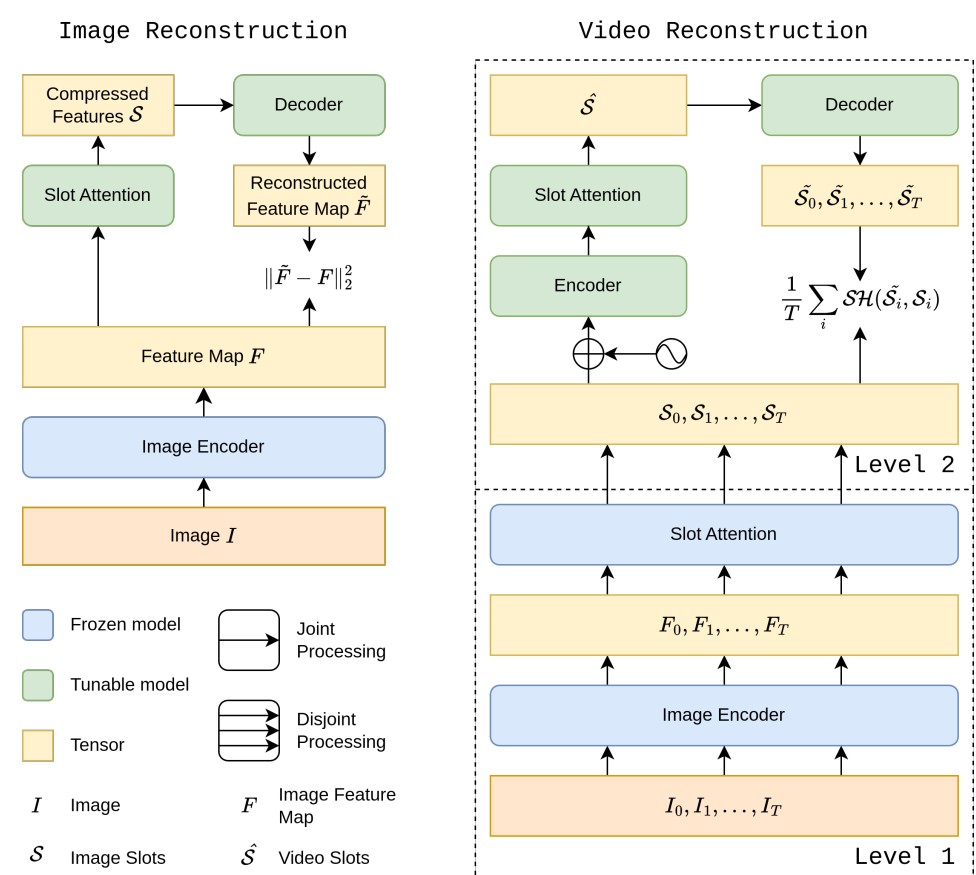

Figure 1: The two stages of `Interpreter`. Blue are frozen modules, green are (hot) tunable modules, yellow are tensors. Single arrows indicate joint-processing, multiple parallel arrows indicate disjoint parallel processing. **Left:** Training of the image-level slot attention auto-encoder by reconstructing image feature maps from a pre-trained encoder. **Right:** Training of the video-level slot attention auto-encoder through reconstruction of image-level object-centric representations.

the decoder maps a set of 2D positionally encoded vectors $P_1 \in \mathbb{R}^{TN \times D_1}$ to reconstructed image-level slots $\tilde{\mathcal{S}}_0 \ldots \tilde{\mathcal{S}}_T$ conditioned on the video-level slots $\hat{\mathcal{S}}$. The objective for this level is measured through the Sinkhorn divergence averaged over time-steps:

$$\mathcal{L}_1 = \frac{1}{T} \sum_i^T \mathcal{SH}(\tilde{\mathcal{S}}_i, \mathcal{S}_i) \qquad (2)$$

**Attention Propagation:** We use the attention maps from the slot attention mechanism for segmentation. Since we employ hierarchical slot attention, there exists no direct attention map between a given feature map $F_i$ and the video-level object-centric representations $\hat{\mathcal{S}}$. Instead, we have to compute this attention map through *attention propagation* between the stages of the hierarchy. More formally, let $A_i \in [0,1]^{HW \times N}$ be the slot attention map representing the linear mapping between the image features $F_i$ and the image-level slots $\mathcal{S}_i$ for frame $i \in [0,T]$, normalized over the slot dimension. Furthermore, let $\hat{A} \in [0,1]^{TN \times K}$ be the slot attention map between the concatenated image-level slots $\mathcal{S}_0||\mathcal{S}_1||...||\mathcal{S}_T$ and video-level slots $\hat{\mathcal{S}}$. Now let $\bar{A} \in [0,1]^{T \times N \times K}$ be the tensor given by decomposing the first dimension of $\hat{A}$ into time and image-slot dimensions. With this, the attention map $M_i$ between $F_i$ and $\hat{\mathcal{S}}$ is given by $M_i = A_i \bar{A}[i]$, where $\bar{A}[i]$ is the $i$:th matrix along the batch dimension of $\bar{A}$.

## 4 EXPERIMENTS

### 4.1 EXPERIMENTAL SETUP

**Datasets:** To evaluate our proposed method on video object segmentation in real-world videos we train and test on YouTube-VIS 2019 (YTVIS-19) (Yang et al., 2019), consisting of a total of 2883 videos that are up to 36 frames, corresponding to 6 seconds of video. We follow the training and testing protocol of (Aydemir et al., 2023) and train on the combined train, test and validation sets. Due to the lack of ground truth masks on the train and validation set, we also follow (Aydemir et al., 2023; Ding et al., 2024) by evaluating on the exact same holdout set of 300 videos from the training set. More succinctly, we train on a total of 2583 videos and evaluate on 300 videos. During training and evaluation we resize the videos and segmentation masks to $336 \times 504$ resolution and extend videos shorter than the maximum length of 36 frames by repeating the sequence to fill the maximum length and truncate any excess. During evaluation we resize the attention maps after propagation with bilinear interpolation to match the resolution of the videos and cut off any repeated frames.

To further test the video segmentation performance we use the synthetic MOVi-E dataset (Greff et al., 2022), consisting of around 10k training videos, and 1k testing videos, at 24 frames each, corresponding to 2 seconds. MOVi-E features up to 20 in-motion objects combined with linear camera motion. We resize the videos to $336 \times 336$ resolution and consume each 24 frame video in full without needing to extend due to constant video length. Similar to YTVIS-19, we use bilinear interpolation to upscale the attention maps during segmentation evaluation. For both YTVIS-19 and MOVi-E we report the video Mean Intersection over Union (mIoU), disregarding the background class. As established by prior methods (Bao et al., 2023; Aydemir et al., 2023; Karazija et al., 2022), we also report the mean per-frame Foreground Adjusted Rand-Index (FG-ARI). Following standard practice, we perform Hungarian matching between the predicted segmentations and the ground truth segmentation maps.

**Implementation Details:** On the YTVIS-19 dataset we train the first level of interpreter with 32 slots and a 4 layer transformer decoder. We keep the dimensionality $D_1$ of the slots equal to that of the 768 dimensional DINOv2 ViT-b embeddings and train with the AdamW (Loshchilov & Hutter, 2018) optimizer for 180 epochs with a batch size of 128, equalling roughly 110k steps. We use a linear learning rate warmup from $1 \times 10^{-4}$ to a peak of $4 \times 10^{-4}$ over 5 epochs, followed by cosine annealing down to $2 \times 10^{-6}$. For the second level we sample 8 slots with dimension $D_2 = 768$ and learn a transformer encoder and decoder, each with 4 layers. We also use AdamW for the second level, but train with a batch size of 96 for 800 epochs, totalling around 21k steps. We use a slow linear learning rate warm up from $1 \times 10^{-5}$ to $2 \times 10^{-4}$ for 50 epochs followed by a cosine decay down to $2 \times 10^{-6}$. Both levels employ a learnable Gaussian prior in the slot attention module, and are trained with a constant weight decay of 0.005. For the second level we set the clustering distance threshold to 0.15, use a blur value of 0.05 (which controls the level of entropy regularization), and set the scaling to 0.5 (which controls the step size of Sinkhorn Knopp). During training of both levels we only use a simple horizontal flip and no other augmentations.

For the MOVi-E dataset we use nearly the same configuration as for YTVIS-19. We sample 32 slots at the first level and 20 at the second level. We train the first level for 60 epochs with a batch size of 128, totalling around 110k training steps, but warm up the learning rate for only 3 epochs. For the second stage we train for 600 epochs with a batch size of 96, totalling around 47k training steps. We warmup for 50 epochs, and perform consine annealing with the same learning rate configuration as with YTVIS-19. We set the clustering distance threshold to 0.15. During training for both level 1 and 2 we use a simple horizontal flip augmentation. Just as with YTVIS-19, we use a blur value of 0.05 and a scaling value of 0.5 for the Sinkhorn loss.

**Ablations:** We perform a small set of ablations on the YTVIS-19 dataset and choose to ablate the number of slots $N$ used at the image level, and the clustering threshold $\epsilon$ used during agglomerative clustering at the video level. Note that we train each model for the distance threshold ablation with half the number of training steps of the final reported model due to resource constraints.

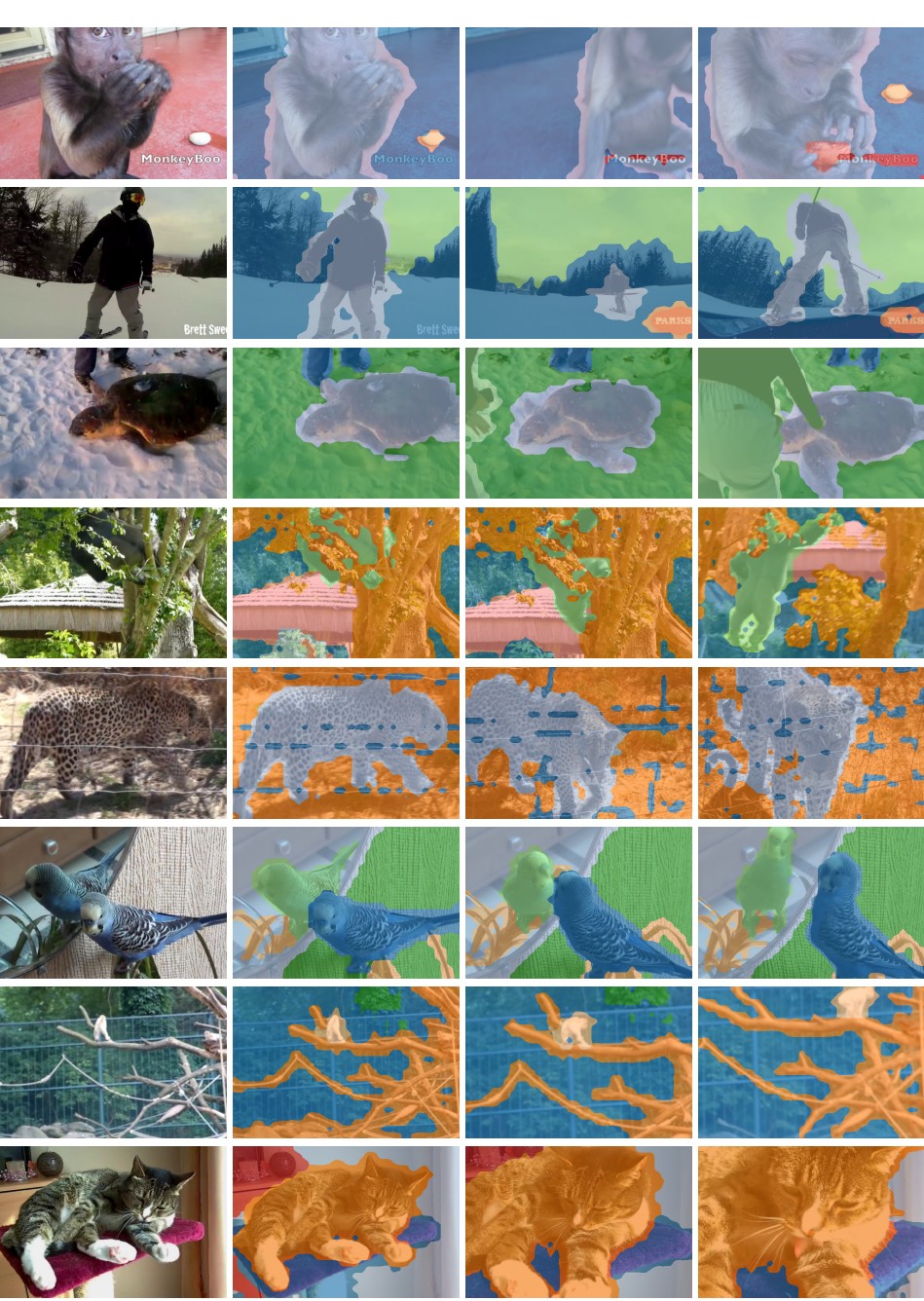

Figure 2: Sample segmentations by `Interpreter` on the 300 video hold-out set from YTVIS-19. Frames were taken from the beginning (first and second column), middle (third column), and end (fourth column) of the videos. First column shows the original RGB frame.

## 4.2 RESULTS

`Interpreter` achieves state-of-the-art results on the YTVIS-19 dataset with a significant 4.1 mIoU improvement compared to the runner up, Betrayed by Attention (Ding et al., 2024), and a 8.9 mIoU improvement over the closest slot attention based method SOLV (Aydemir et al., 2023) (table 1). However, the method lags behind on image FG-ARI, indicating that the per-frame segmentation could be improved. We demonstrate some segmentation samples from the 300 testing videos in figure 2. `Interpreter` is able to deal with occlusions, as seen in the first row of figure 2, where a pebble is occluded for a prolonged period. It is also able to track subjects that see significant changes in appearance, size, pose and position, as seen in the second video of a skier. Entities going from off-screen to on-screen are also segmented well, as seen in both first and and third row, where a fruit and person are revealed, respectively. The fourth row shows a particularly impressive performance, identifying a heavily occluded gorilla hidden in a tree that later makes a leap of faith. In the fifth row we see the model dealing well with segmenting the leopard behind the thin-wired fence. The sixth and seventh row shows `Interpreter` managing fine-detailed segmentations effectively, segmenting the thin leaves and branches, respectively, with remarkable accuracy. The small monkey is also tracked well in row seven. The last row shows a cute cat.

Table 1: Performance on YTVIS-19 in terms of FG-ARI and mIoU. For TimeT (Salehi et al., 2023) we borrow the rerun evaluations from (Ding et al., 2024). For our proposed method we rerun the evaluation three times with random seeds and report the mean.

| Model | FG-ARI | mIoU |
|---|---|---|
| SAVi (Elsayed et al., 2022) | 11.1 | 12.7 |
| STEVE (Singh et al., 2022) | 20.0 | 20.9 |
| OCLR (Xie et al., 2022) | 15.9 | 32.5 |
| VideoSAUR (Zadaianchuk et al., 2023) | 39.4 | 29.1 |
| SOLV (Aydemir et al., 2023) | 29.1 | 45.3 |
| SMTC (Qian et al., 2023) | 31.4 | 38.8 |
| TimeT (Salehi et al., 2023) | 37.9 | 40.4 |
| BA (Ding et al., 2024) | **44.3** | 50.1 |
| `Interpreter` (ours) | 28.5 | **54.2** |

Table 2: Performance on MOVi-E in terms of FG-ARI and mIoU. For our proposed method we rerun the evaluation three times with random seeds and report the mean.

| Model | FG-ARI | mIoU |
|---|---|---|
| SAVi (Elsayed et al., 2022) | 42.8 | 16.0 |
| STEVE (Singh et al., 2022) | 50.6 | 26.6 |
| VideoSAUR (Zadaianchuk et al., 2023) | 73.9 | 35.6 |
| SOLV (Aydemir et al., 2023) | 80.8 | - |
| BA (Ding et al., 2024) | 84.4 | **40.7** |
| `Interpreter` (ours) | **85.3** | 29.7 |

On the MOVi-E dataset we see that our proposed method falls behind previous works (-11 mIoU, table 2). This is an interesting observation, as we achieve good performance on YTVIS-19. We look closer at why our method appears to perform worse on MOVi-E in subsection 4.3. Surprisingly, our method achieves the highest FG-ARI on this dataset.

Looking at the ablation results in table 3a we see a significant rise in mIoU as the number of image-level slots $N$ increases, achieving an mIoU of 54.2 at 32 slots. However, FG-ARI does not portray such a trend, achieving the best performance at 16 image-level slots. Optimizing for mIoU we train our final model on YTVIS with 32 image-level slots. Varying the clustering threshold $\epsilon$ has a rather pronounced effect on both FG-ARI and mIoU (see table 3b), achieving an increase of most 3.3 FG-ARI and 4.8 mIoU when compared to having no clustering. Optimizing for mIoU performance we set $\epsilon = 0.15$ on the final YTVIS-19 model.

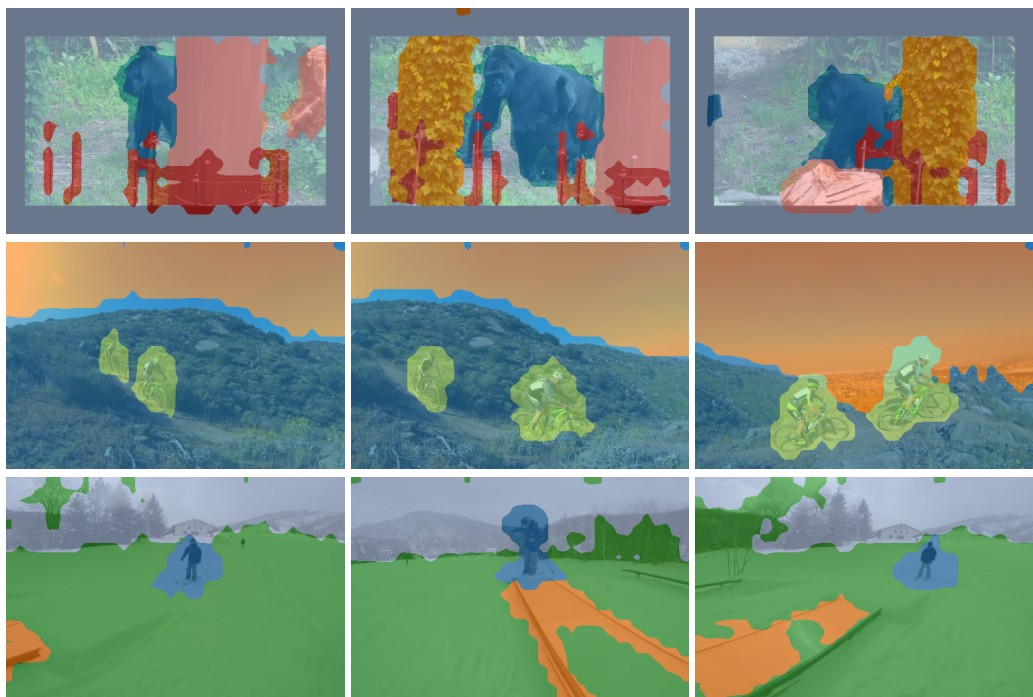

Figure 3: Sample failure cases by `Interpreter` on YTVIS-19. **Top:** we can see how an adult gorilla and the baby hanging on to it are clustered together. **Middle:** we see how two bikers are consistently clustered together throughout a video. **Bottom:** shows rapid camera motion and subsequent confusion of two entities.

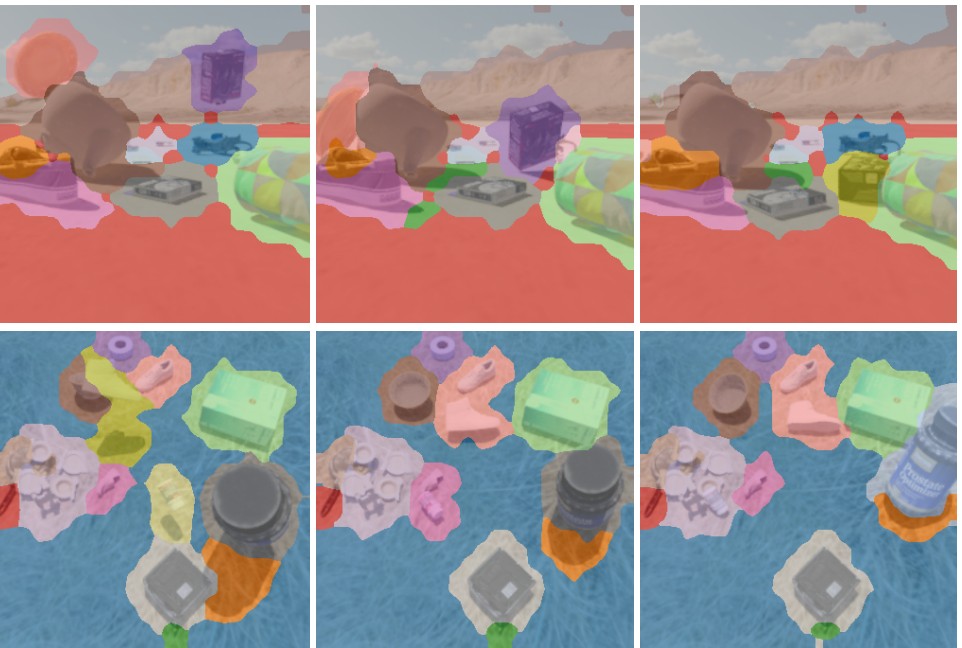

Figure 4: Sample failure cases by `Interpreter` on the MOVi-E test set. **First row:** take note of how the flying box changes slot assignments when it lands on the ground, going from purple to yellow. **Second row:** the pill bottle changes assignment between flying and landing on the ground. We can also see the spurious clustering of two shoes at the top of the video.

Table 3: Ablation studies on segmentation performance with varying parameters. We report the mean over 3 random seed evaluations.

(a) Segmentation performance on YTVIS-19 as the number of image-level slots $N$ varies.

| $N$ | FG-ARI | mIoU |
|-----|--------|------|
| 8 | 28.0 | 49.3 |
| 16 | **29.3** | 52.0 |
| 32 | 28.5 | **54.2** |

(b) Segmentation performance on YTVIS-19 as the clustering distance threshold varies.

| $\epsilon$ | FG-ARI | mIoU |
|------|--------|------|
| 0.00 | 26.0 | 49.2 |
| 0.05 | 27.8 | 50.5 |
| 0.10 | 28.3 | 53.5 |
| 0.15 | 28.4 | **54.0** |
| 0.20 | **29.3** | 53.7 |

## 4.3 ANALYSIS OF RESULTS

We observe a few common failure modes of our proposed method. A common issue on the YTVIS-19 dataset is the spurious clustering of similar entities. This usually presents itself when there are two or more objects of similar appearance, and seems to happen more frequently, though not exclusively, when such objects are in close proximity. Figure 3 shows two examples of this happening on the 300 video YTVIS-19 hold-out set. Another failure mode occurs when there are similar entities and significant camera movement. In particular, in cases of rapid camera movement can cause the model to confuse one entity for another due to camera panning. We show an example of this in the bottom row of figure 3.

Looking closer at the qualitative results in figure 4 we can see one of the reasons why the performance suffers on MOVi-E. Ostensibly, the model appears to allocate excess slot capacity at the video level to break up the movement trajectories of objects. Notably, we can see that objects maintain the same slot assignment as they fall through the air, but change assignment as soon as the object hits the ground and motion changes. This is unsurprising considering the compression objective; the model is making use of the allotted representational capacity to best reconstruct the sequence of image-level object-centric representations. To remedy this the clustering distance threshold $\epsilon$ could be increased, but this also makes spurious clusterings of adjacent objects more likely. Notably, we observe qualitatively that MOVi-E is sensitive to under-clustering as $\epsilon$ increases, likely due to the sheer number of objects in each scene.

The attention maps derived from slot attention have a tendency to over-segment objects, particularly when the background is simple, leading to a signature "aura" look. An example of this can be observed in the middle row in figure 2. This might be why our proposed method, which makes use of slot attention, falls behind other works such as BA and TimeT (Ding et al., 2024; Salehi et al., 2023), that do not make use of slot attention, in terms of average per-frame FG-ARI on YTVIS-19. Supporting this hypothesis is the relatively close performance of prior slot-attention based methods; SMTC, SOLV, and our proposed method score closely at 31.4, 29.1, 28.5 FG-ARI, respectively. On MOVi-E `Interpreter` scores unexpectedly well in terms of per-frame FG-ARI. This result is consistent with the observations of motion trajectories being segmented into separate parts; while the video-level mIoU score will suffer from segmented motion, the per-frame FG-ARI results should not. Regardless, the performance in terms of FG-ARI on MOVi-E is surprising.

## 5 CONCLUSION

### 5.1 SUMMARY

We have introduced `Interpreter`, a novel method for efficient video-level representation learning that leverages a hierarchical slot attention architecture with local learning objectives. By building upon pre-trained image backbones and avoiding the reconstruction of full frame feature maps, `Interpreter` scales effectively to longer videos without the need for chunking and matching. We enable this novel hierarchical approach by introducing a method for propagating attention maps between slot attention layers and a method for learning to reconstruct unordered sets of object-centric representations using the Sinkhorn divergence distance metric.

Experimental results demonstrate that `Interpreter` achieves state-of-the-art performance on the YTVIS-19 dataset in terms of mean Intersection over Union, outperforming existing methods by a significant margin. While there are limitations observed on synthetic datasets like MOVi-E, where the method underperforms compared to previous works, `Interpreter` represents a notable advancement in object-centric video representation learning as it offers a scalable and efficient approach for processing entire videos.

## 5.2 FUTURE WORK

We see a need for better methods of dynamically selecting the number of slots used during training and inference time. The agglomerative clustering approach helps to alleviate over-clustering in some scenarios, but it can lead under-clustering in others. A better method of deciding the number of slots would likely improve our proposed method.

We empirically find that `Interpreter` is particularly sensitive to masking. During our experiments we tried temporal masking for when videos are too short to fill the context window, such as is the case with YTVIS-19, but found that it often lead to divergence during training. While it is possible to mask, we found that extending the sequence through repetition to be an easier solution that does not adversely affect results. A future direction of research could try to address this issue.

The hierarchical slot attention method we propose is general and could be extended to any depth. It would be interesting to see it applied with more levels than two, which would allow the method to be extended to even longer videos and would result in multi-level segmentation hierarchies.

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
