# OpenReview forum: "Efficient Object-Centric Learning for Videos"
_ICLR.cc/2025/Conference — ICLR 2025 Conference Withdrawn Submission_

### Official Review · Reviewer_ipv7 · 2024-11-02

**Soundness:** 2
**Presentation:** 1
**Contribution:** 1
**Rating:** 3
**Confidence:** 5

**Summary:**

This paper introduces a hierarchical slot attention approach for handling temporal context in video segmentation. To achieve this, it incorporates a video-level slot that aggregates temporal information across all frame-level slots. Additionally, to smoothly apply the video-level slot for video representation prediction, the paper proposes an attention map propagation technique. For loss calculation, Sinkhorn Divergence is utilized. With these components, the proposed model, Interpreter, achieves state-of-the-art performance on the YTVIS-19 dataset in terms of mIoU and on the MOVi-E dataset in terms of FG-ARI.

**Strengths:**

The paper introduces a straightforward design for unsupervised video object segmentation using slot attention. This architecture demonstrates remarkable performance on the real-world dataset YTVIS-19.

**Weaknesses:**

**1. Limited Architectural Contribution**

The primary contribution of Interpreter lies in its hierarchical architecture, a concept previously introduced in the video instance segmentation method, VITA [1]. Similar to VITA, Interpreter employs a hierarchical design where video-level queries aggregate temporal context from frame-level queries. Apart from differences in target tasks and objective functions, the overall architectural design remains largely similar to that of VITA.

**2. Insufficient Experimental Support**

Additional experiments are necessary to validate the proposed methods. In Tables 1 and 2, as noted by the authors, results on the MOVi-E dataset show a trend that significantly deviates from results on YTVIS-19. Section 4.3 discusses specific cases with limited examples to interpret these discrepancies. However, since these two sets of results exhibit opposite trends, it remains challenging to conclude that the proposed method is generally applicable. To address these contradictions, further analysis—such as statistical investigation—would be beneficial. Additionally, only two ablation studies are presented, even for critical hyperparameters, and key factors like the effect of varying the number  K  are not explored.

**3. Limited Readability**

The overall structure of the paper hinders readability and comprehension. In particular, the experimental section is challenging to follow, as it combines main experimental results, qualitative findings, and ablation studies within the same section, making it difficult to discern the purpose and implications of each individual experiment. Furthermore, Figures 3 and 4 display segmentation results without the original samples, which complicates the reader’s ability to fully interpret the analysis.

[1] Heo, Miran, et al. "Vita: Video instance segmentation via object token association." Advances in Neural Information Processing Systems 35 (2022): 23109-23120.

**Questions:**

**Q1. What are the main architectural differences between Interpreter and VITA?**

In terms of architectural design, what distinguishes Interpreter from VITA? Could you specify the unique aspects of Interpreter’s approach, especially in how it addresses hierarchical design and temporal context aggregation?

**Q2. Is there any statistical basis for the analysis beyond observations on a few samples?**

Beyond observations from a limited set of examples, does the paper offer any statistical foundation for its analysis? For instance, is there evidence that specific factors like object movement or motion changes hinder the performance of slot-based approaches? A more comprehensive investigation into such cases could help substantiate the findings.

**Q3. Are additional ablation studies provided to validate the proposed method’s effectiveness?**

Apart from the current experiments, are there further ablation studies examining key factors, such as the influence of varying the number of slots (K) or the impact of end-to-end fine-tuning in the second stage?

---

### Official Review · Reviewer_jWcK · 2024-11-02

**Soundness:** 3
**Presentation:** 3
**Contribution:** 2
**Rating:** 5
**Confidence:** 4

**Summary:**

The work introduces an unsupervised approach to object-based segmentation of video sequences. The approach comprises two stages. The first stage follows previous work and trains an autoencoder that decomposes an input image into a set of slot tokens. In the second stage, another autoencoder learns to represent the set of slot tokens, extracted from each frame in a video, with a more compact set of video-level slot tokens. To train this autoencoder, the approach leverages Sinkhorn divergence, which establishes a (relaxed and differentiable) correspondence between the set of predicted tokens and the input set. The final output -- a temporally consistent segmentation -- is the result of attention propagation, which relies on the similarity between image-specific slots and the video-level slots. The results on YouTube-VOS and synthetic MOVi-E demonstrate impressive segmentation quality, but the quantitative results are a bit mixed.

**Strengths:**

* I like the work’s technical contribution, the Sinkhorn divergence. However, I’d encourage the authors to include more details (what’s behind SH function in (2)).
* The approach is technically sound. It makes a lot of sense to represent a video with a compact set of slot tokens and computing the segmentation through attention propagation, as described in ll. 206-215.
* Fig. 1 provides a great overview for the approach, which helps in following the technical details. (Remark: It could’ve been more compact and use vectorised graphics).
* I enjoyed that the text does not stop after the mixed quantiative results, but instead makes a good effort to analyse and explain them.

**Weaknesses:**

* The exposition, especially the technical part, feels way to congested. I would have preferred more technical details in Sec. 3.2 than the  Figures 2-4 loading two full pages, which feel a bit like space-fillers. For example, the work does not really explain how the Sinkhorn divergence is computed in Eq. (2), nor does it really explain the architecture of the encoders/decoders in the two stages of training, etc.
* The results are obviously mixed: On YouTube-VOS the approach discriminates between the objects well, but falls behind on foreground-background segmentation and vice-versa on MOVi-E. I like that the text discusses these weaknesses, but the analysis would have been more convincing with more informative qualitative examples (including the ground truth and the output from previous work).
* The title falls short on the promise of efficiency. Perhaps the method is efficient, but I did not find convincing arguments or corresponding experiments to support this point.
* The experiments are bit too brief. I would be curious to see the approach with another pre-trained backbone and dataset (e.g. DAVIS).

**Questions:**

* How is the Interpreter more efficient tham previous work? (e.g. STEVE, BA, VideoSAUR).
* How does the model compare to previous work if normalised for the pre-trained architecture? E.g. BA uses ViT-s/8, while ViT-B/14 is used here.
* Interpreter is developed with long videos in mind. What is the definition of “long” in this work and how is this reflected in the experimental setup?
* How would approach compare to move naive objectives, e.g. matching the slots with Hungarian matching and minimising the corresponding distance (e.g. L1/L2)?

---

### Official Review · Reviewer_cf9z · 2024-11-03

**Soundness:** 2
**Presentation:** 2
**Contribution:** 1
**Rating:** 3
**Confidence:** 5

**Summary:**

The paper presents a method called Interpreter, aimed at efficient, unsupervised video-level object-centric representation learning. Interpreter introduces a hierarchical slot attention architecture where image-level representations are compressed first, then video-level representations are derived from them using a relaxed optimal transport objective, Sinkhorn Divergence, for unsupervised segmentation. This approach circumvents the typical computational load associated with reconstructing frame-level feature maps, allowing Interpreter to process longer videos effectively. Experiments show that Interpreter achieves strong results on the YTVIS-19 dataset and synthetic datasets like MOVi-E.

**Strengths:**

1. The paper is well-written and structured, with clear explanations of the novel hierarchical slot attention mechanism and its advantages in scaling object-centric representation to longer videos.

2. The approach of per-frame slot attention followed by video-level slot attention is both novel and elegant, allowing the model to handle temporal dependencies across the entire video without chunking.

**Weaknesses:**

1. The second-level slot number does not stay under ten (8 for YTVIS-19), which contradicts the paper’s claim of handling extensive temporal context effectively (L101).

2. Results on the DAVIS-17-unsupervised dataset are absent, and performance on metrics (FG-ARI and mIOU) shows considerable variation across different benchmarks, suggesting limitations in the model’s generalizability.

3. The discussion around FG-ARI and mIoU metrics lacks sufficient depth, especially in explaining the model’s inability to perform consistently across both benchmarks. It remains unclear why the method does not yield strong outputs on both metrics concurrently​.

**Questions:**

1. Could the authors clarify the significant discrepancy observed between FG-ARI and mIoU performance in this model? In my view, both FG-ARI and mIoU should be high if object segmentation remains accurate over time.

2. Have the authors considered using only frame-wise slot representations at the second level (where the same slot index per frame corresponds to the same object), rather than applying slot attention at the video level? What would be the implications of this approach?

3. To what extent is the DINOv2 feature extractor crucial for this model? Would the method fail without it?

4. Why is a different number of second-level slots used for the YTVIS-19 and MOVi-E datasets?

---

### Official Review · Reviewer_CfWA · 2024-11-04

**Soundness:** 1
**Presentation:** 1
**Contribution:** 2
**Rating:** 1
**Confidence:** 3

**Summary:**

The authors present Interpreter, a VOS method based on hierarchical slot attention that consists of separate image-level and video-level processing. To compute the image-level attention slots, Interpreter uses implicit slot attention to learn object-centric features from an image-trained backbone. Implicit slot attention is also used at the video level to learn object representations across frames, relying on the Sinkhorn divergence to learn correspondence between sets of slots across different frames. Experiments are conducted on the YTVIS-19 and MOVi-E datasets to compare Interpreter to other slot-attention-based methods. An ablation study is carried out on YTVIS-19 to determine the effect of number of slots and clustering distance threshold.

**Strengths:**

1. Significant qualitative results are included, including failure cases.
2. The technical contribution appears to be novel for VOS.

**Weaknesses:**

1. Issues with Experimental Evaluation.

    a. The paper claims Interpreter is a VOS method but performs its evaluation using a Video Instance Segmentation (YTVIS-19) and Video Semantic Segmentation dataset (MOVi-E). If Interpreter is a VOS method then it should be evaluated on VOS datasets such as DAVIS [1], and compared to the state of the art VOS methods, in order to assess the contribution of the work.

    b. The paper claims Interpreter targets long videos but the length of the videos in the datasets chosen are on the order of seconds not minutes, making it difficult to verify this claim.

    c. Qualitative results are included for Interpreter but not for competing methods, making it difficult to assess the performance quality of Interpreter.

2. Exposition of method lacks mathematic details. In particular, Sinkhorn Divergence is never defined mathematically and the final loss function is not included. This makes it difficult to understand the method beyond a surface level.

3. Related works section lacks mention of query-key-value retrieval-based methods such as STM [2] for VOS, which is a major and important direction for the task. The motivation for using slot attention based methods is not clear.

4. Writing is not direct. For example, it should be explained why Interpreter performs "unexpectedly well" in l. 471 and what is "surprising" in l. 474. As another example, the phrase in l. 339 "The last row shows a cute cat." seems out of context.

[1] "The 2017 DAVIS Challenge on Video Object Segmentation". J. Pont-Tuset, F. Perazzi, S. Caelles, P. Arbeláez, A. Sorkine-Hornung, and L. Van Gool. arXiv:1704.00675, 2017.
[2] "Video Object Segmentation using Space-Time Memory Networks". Seoung Wug Oh, Joon-Young Lee, Ning Xu, Seon Joo Kim. ICCV, 2019.

**Questions:**

Please address the points brought up in the weaknesses above.

---

### Author Response · Authors · 2024-11-25

We kindly thank the reviewers for providing their valuable feedback.

After reviewing the provided criticisms, we agree that the paper needs more work. In particular, the paper should provide a more rigorous exposition of Sinkhorn, model architecture and training, and a better motivation for the methods efficiency. Additionally, more ablations (e.g. testing different backbones), and quantitative results should be added to strengthen the existing mixed results, with a more thorough investigation into the proposed method's points of failure. Further comparisons with existing methods should also be added, in addition to more clarity showing how Interpreter is distinguished from prior works.

We once again thank the reviewers for their valuable and thorough feedback, and for highlighting both the strengths and weaknesses of our submission.

---

### Note · Authors · 2024-11-25

I have read and agree with the venue's withdrawal policy on behalf of myself and my co-authors.